# Extension to the Jiles–Atherton Hysteresis Model Using Gaussian Distributed Parameters for Quenched and Tempered Engineering Steels

**DOI:** 10.3390/s25051328

**Published:** 2025-02-21

**Authors:** Alasdair Regan, John Wilson, Anthony J. Peyton

**Affiliations:** Department of Electrical and Electronic Engineering, The University of Manchester, Oxford Road, Manchester M13 9PL, UKjohn.wilson@manchester.ac.uk (J.W.)

**Keywords:** Jiles–Atherton, hysteresis, ferromagnetism, non-destructive testing

## Abstract

The Jiles–Atherton (J–A) model has seen extensive use for modelling the hysteresis behaviour of ferromagnetic materials due to its computational efficiency, simplicity of use, and small number of physically related parameters. However, in this work, the application of the J–A model to hysteresis curves obtained from experimental measurements for as-quenched and quenched-and-tempered engineering steels is considered. It has been demonstrated that the current form of the J–A model is not capable of representing certain observed features in the obtained hysteresis curves of these steels, in particular, the rapid narrowing of the loops seen for as-quenched steels and the sharp corners seen for quenched-and-tempered steels. This work has shown that a superior fit to the major loops for such steels can be obtained by applying Gaussian variations with respect to the applied magnetic field to the model parameters. The findings are supported by experimental results from engineering steels used in the oil and gas industry.

## 1. Introduction

Modelling of the hysteresis properties of magnetic materials is complex, and a number of hysteresis models have been proposed in the literature. Additionally, a number of review papers have been published [1,2,3], though the authors are not aware of a single comprehensive source. One such model that has seen widespread application is the Jiles–Atherton (J–A) model [4,5,6]. The J–A model is one of a number of macroscopic models of ferromagnetic magnetisation and hysteresis. While macroscopic magnetisation models lack the rigorous physical basis of micromagnetic models, their comparative simplicity makes them appropriate for applications where the macroscopic response of materials to external fields is of interest. Particularly, attractive features of the J–A model model are the use of a small number of physically related and measurable parameters, reasonable computational efficiency, and simplicity of use [1,7]. The model has been widely applied to real engineering problems and is used to implement ferromagnetic hysteresis in a number of commercial finite element modelling packages, such as COMSOL [8] and Altair [9].

This model has seen a huge amount of practical use, likely due to its ability to closely model the behaviour of common electrical steels, which, due to their applications in electrical machinery and transformer accounts for a significant fraction of materials considered when modelling magnetic materials. However, the model’s ability to model the magnetic behaviour of high-performance ferritic engineering steels is often poor, and this limits the potential of the model for use in the development of models for magnetic systems incorporating these materials, such as sensors for magnetic non-destructive testing.

This work presents a modified form of the J–A model that permits variation of the model parameters in order to better represent common types of engineering steel of interest in the field of non-destructive testing. Experimental measurements were conducted for a selection of oil and gas steel samples and these were used to verify the efficacy of the proposed model.

## 2. The Jiles–Atherton Model

The 1D J–A model represents hysteresis using a differential equation that must be solved numerically. The core equation of this model is given by Equation (Equation 1) [7].(1)$$dM=\left\{\begin{array}{cc} \frac{1}{k\delta }\left[\left(M_{an}-M\right)dH_{e}\right]+cdM_{an}, & \left(M_{an}-M\right)dH_{e}>0\\ cdM_{an}, & \left(M_{an}-M\right)dH_{e}\leq 0 \end{array}\right.$$

In Equation (Equation 1), *M* is the magnetisation, $H_{e}$ is the effective applied field, and $M_{an}$ is the anhysteretic magnetisation. *k* and *c* are two of the five model parameters known as the pinning parameter and the reversibility parameter, respectively. δ is a directional parameter given by Equation (Equation 2).(2)$$\delta =\left\{\begin{array}{cc} +1, & \frac{dH}{dt}>0\\ -1, & \frac{dH}{dt}<0 \end{array}\right.$$

The effective field is taken from mean field theory and is given by the Expression (Equation 3), where α is another model parameter representing interdomain coupling.(3)$$H_{e}=H+\alpha M$$

The anyhysteretic magnetisation is typically given by a modified Langevin function:(4)$$M_{an}=M_{s}\left[\coth\frac{H_{e}}{a}-\frac{a}{H_{e}}\right]$$
where the remaining two model parameters, $M_{s}$ and *a* represent the saturation magnetisation and what is known as the Langevin slope parameter, respectively.

The J–A model is stable so long as Equation (Equation 5) is satisfied [10].(5)$$M_{s}\alpha <3a$$

It is not possible to obtain an analytical solution to the differential Equation (Equation 1), and numerical integration techniques must be used to obtain a solution [11,12,13].

## 3. The Modified Jiles–Atherton Model

It has been found in this study that the J–A model cannot accurately represent the magnetic properties of some of the harder mechanical steels of interest in non-destructive testing applications. This problem has been found to be particularly apparent for as-quenched steels with martensitic microstructures and quenched and tempered steels with tempered martensite microstructures.

There has been little discussion of the application of the J–A model to engineering steels in the literature; however, there have been some examples of application to magnetically hard permanent magnet materials. In some cases, reasonable results were obtained [14,15,16]. Gao et al. [17] found for NdFeB permanent magnets that the model could not accurately model the long, almost reversible approach to saturation observed with this material. This result is similar to what is seen for as-quenched steels. Brachtendorf and Laur [18] proposed an alternative J–A based model in order to better represent hard magnetic materials with very square hysteresis loops by replacement of the anhysteretic function. Annakkage et al. [19] proposed a similar approach to better fit the shoulder region B-H curves obtained from a current transformer core with a relatively square B-H curve.

### 3.1. Proposed Model Alterations

The proposed modifications to the model are an extension of the modification made by Toman et al. [20] to the pinning parameter *k*, where a Gaussian variation was applied to reduce the parameter at higher applied fields. The variation to *k* is retained in the proposed model, and additional Gaussian variations are applied to the other four model parameters. The modification to these parameters increases the value of the parameter at higher fields.

The J–A model is based on a relatively simple representation of the physics of magnetic materials and does not consider many of the complex factors impacting the magnetic behaviour of materials. In reality, the magnetic behaviour of the material is influenced by complex microstructural features including, but not limited to, phase composition, dislocation density, crystallographic texture, and grain size. As an external field is applied, the magnetic domains in the material will either grow, shrink, or rotate to align with the local effective field. As these domain processes evolve, they will interact with these microstructural features, and these features may influence the interdomain effects. As such, it is unreasonable to expect the parameters governing the evolution of magnetisation in materials to be constant over the full range of the hysteresis curve. The modifications presented here are motivated by providing a method by which these parameters can vary to better capture the observed physical behaviour of the magnetic materials under consideration.

In choosing functions to represent the variation of these parameters in a manner that is suitable for the practical representation of the magnetic behaviour of engineering steels, it is important to consider both the suitability of the function for numerical modelling and the efficacy of the functions in representing the observed physical behaviour. The use of Gaussian functions for the parameter variations is numerically appropriate since they vary smoothly and have no discontinuity around zero. In this work, it is shown empirically that the proposed Gaussian variations are effective in representing the observed magnetic behaviour of the steels under consideration. While the proposed form of these parameters does not have a rigorous physical justification, Gaussian variations are not an unreasonable choice if one considers the statistical distribution of low-field type magnetic interactions to high-field type interactions occurring at a given applied field.

For each model parameter, the modified form of the parameter is dependent upon *H* with a Gaussian-type variation and has three fitting parameters: $p_{0}$, $p_{1}$, and $\sigma_{p}$. $p_{0}$ and $p_{1}$ control the magnitude of the fixed and variable parts of the parameter, while $\sigma_{p}$ controls the shape of the Gaussian function. It was found that modified forms for all five model parameters were required to accurately fit to the curves under consideration.

#### 3.1.1. Parameter: *a*


(6)
$$a=a_{0}+a_{1}\left(1-\exp(-\frac{H^{2}}{2\sigma_{a}^{2}})\right)$$


This parameter predominantly affects the slope in the central section of the anhysteretic curve. It was found that allowing this parameter to vary along with the other model parameters facilitated a better fit to the measured data.

#### 3.1.2. Parameter: α


(7)
$$\alpha =\alpha_{0}+\alpha_{1}\left(1-\exp(-\frac{H^{2}}{2\sigma_{\alpha }^{2}})\right)$$


This parameter has a similar effect to *a*, though it is more subtle. As with *a*, it was found that allowing α to vary along with the other model parameters facilitated a better fit to the measured data.

#### 3.1.3. Parameter: *c*


(8)
$$c=c_{0}+c_{1}\left(1-\exp(-\frac{H^{2}}{2\sigma_{c}^{2}})\right)$$


As irreversible pinning processes reduce at higher fields and reversible domain rotation processes begin to dominate, it is appropriate to increase *c*, which represents the magnitude of reversible magnetisation processes at high fields.

#### 3.1.4. Parameter: *k*


(9)
$$k=k_{0}+k_{1}\exp\left(-\frac{H^{2}}{2\sigma_{k}^{2}}\right)$$


The form of *k* is unchanged from that proposed by Toman [20], where a Gaussian variation was applied to reduce the parameter to a lower fixed value at high fields. A physical justification for the use of this function can be made by considering that the *k* parameter controls the loss due to irreversible magnetisation processes, typically pinning. Pinning processes dominate at lower applied fields; however, as the material magnetises, the magnetic domains grow, and domain walls start to disappear, leading to domain rotation processes dominating at higher fields.

#### 3.1.5. Parameter: $M_{s}$


(10)
$$M_{s}=M_{s0}+M_{s1}\left(1-\exp(-\frac{H^{2}}{2\sigma_{M_{s}}^{2}})\right)$$


$M_{s}$ represents the saturation magnetisation and, as such, may seem a strange parameter to vary; however, the standard J–A cannot represent the long sloped approach to saturation seen in many engineering steels. This process is likely a consequence of domain rotation, which is not considered in the derivation of the model. Allowing the saturation magnetisation to vary in the manner proposed allows for this high-field behaviour to be represented.

## 4. Materials and Methods

The major loop B-H characteristics of several samples of oil and gas steels were experimentally determined, and the proposed modified J–A model was fitted to these measurements to determine the efficacy of the proposed model. Details of the samples used and the experimental and computational methods are detailed below.

### 4.1. Details of Samples

The samples used for this work were machined from a pair of oil and gas steels subjected to a range of heat treatment processes. Details of the samples are given in Table 1. Hardness was measured using the Rockwell C hardness scale, with the exception of sample D2, which is a softer steel; as such, a reliable measurement on the C scale was not available, and the Rockwell B hardness scale was used instead.

#### 4.1.1. Steel Chemistries

Two steel chemistries were provided: one steel was a chromium-molybdenum Cr-Mo steel, and the other a carbon-manganese *C*-Mn. In both cases, the steels had additions of boron and titanium. As the steels are proprietary products, exact chemical compositions and specific end applications were not provided.

#### 4.1.2. Heat Treatments

The heat treatment processes mentioned above were chosen to produce a set of samples for both of the two steels with the following properties:A: As quenched samples with a fully martensitic microstructure.B: Quenched and tempered samples with a hardness in the region of 20 HRC.C: Quenched and tempered samples with a hardness in the region of 30 HRC.D: Normalised samples with a ferrite-pearlite-bainite microstructure.

Of these samples, the quenched and tempered samples were representative of end products; the as-quenched samples and normalised samples were representative of intermediate heat treatment stages in the manufacturing process.

### 4.2. Experimental Method

The B-H curves considered in this work were obtained using the following method.

A diagram of the measurement system developed for B-H measurement is shown in Figure 1.

A low-frequency time-varying signal was fed to a bipolar power amplifier, which supplied sinusoidal current to two series-connected 300-turn excitation coils constructed from 0.7 mm diameter enamelled copper wire. The coils were wrapped around a 26 × 36 mm laminated silicon-steel core. The cylindrical sample to be tested was fitted into a semi-circular slot in the core to maximise coupling between the core and the sample.

The axial applied field (H) was measured using a quantum well Hall sensor developed at the University of Manchester [21] on the surface of the sample. The Hall sensor was positioned in the centre of the sample, equidistant between the poles of the electromagnet. This surface measurement was used to infer the internal applied field by exploiting the fact that the tangential component of the applied field is continuous across a boundary in the absence of surface currents. The low excitation frequency ensured that any surface currents were negligible.

The induced field (B) was measured using a 100-turn encircling coil constructed with 0.4 mm diameter enamelled copper wire.

It is noted that the geometry of this measurement system cannot be regarded as fully closed-circuit. However, the resultant geometric influence on the obtained measurements has not been found to significantly alter the results when compared to commercially available closed-circuit systems, as long as the magnitude of the applied field does not significantly exceed 20 kA/m. The system provides a repeatable method for obtaining comparative measurements from the often small samples available for engineering steels, such as those considered in this study.

#### 4.2.1. Calculating the B-Field

The flux density, $B_{0}$, is calculated by integrating the voltage, *v*, induced across the search coil.(11)$$B_{0}=\int \frac{v}{A_{s}N}dt$$
where $A_{s}$ is the cross-sectional area of the sample and *N* is the number of turns on the search coil.

In order to compensate for the additional flux through the search coil due to the difference in cross-section of the sample and coil, a compensation term is added to find the flux density in the sample, *B*.(12)$$B=B_{0}-\left[\left(\frac{A_{w}-A_{s}}{A_{s}}\right)\mu_{0}H\right]$$
where $A_{w}$ is the cross-sectional area of the search coil and *H* is the applied field.

Typically, the flux coil voltage, *v*, obtained experimentally will contain some degree of DC offset, which results in a linear drift of the B-field values obtained after integration. This is dealt with by detrending the signal after integration.

#### 4.2.2. Sample Preparation

Samples were prepared by machining the material into uniform cylinders with a diameter of 5 mm and a length of 50 mm.

The samples were cut using wire erosion to minimise heating of the sample and the associated risk of microstructural changes and turned to remove the wire on/off pip.

#### 4.2.3. Demagnetising the Sample

Before taking each measurement, it is important to demagnetise the sample. This ensures all measurements start from the same state and ensures the integration constant arising during the evaluations of (Equation 11) is zero.

This is done by applying an alternating field that decreases over time from the peak field used to zero. A frequency of 10 Hz was used for the demagnetising cycles.

#### 4.2.4. Major Loop Measurement

Major B-H loops are the hysteresis loops generated by applying a symmetric alternating field of sufficient magnitude to bring the material to technical saturation. The resultant loop shows the path taken in reversing the direction of the magnetisation from one saturated state to another.

Major B-H loops were measured for all samples using a sinusoidal 10 mHz excitation with the apparatus described above. Such a low frequency was used to reduce eddy currents in the sample. Each sample was demagnetised before each measurement, and several cycles of each loop were averaged to reduce noise.

### 4.3. Computational Methods and Parameter Fitting

A number of methods have been proposed for fitting the model parameters to experimental measurements. Early attempts typically relied on tuning the model parameters to a set of reference points on the measured curves [22,23,24], but such methods have largely been superseded by iterative optimisation methods. Such methods include the application of global optimisation methods, such as Simulated Annealing [25], Genetic Algorithms [26,27], or Differential Evolution [12,20], directly to experimental measurements. Local optimisation methods can also be applied to rough curve fits obtained using either reference point methods [28,29,30] or global optimisation [31,32]. Trapanese [33] demonstrated the potential of neural networks for parameter fitting of the J–A model.

For this work, the differential evolution approach proposed by Toman [20] was found to be successful. In this method, the objective function given by Equation (Equation 13) is minimised using differential evolution.(13)$$\epsilon =\sqrt{\sum_{i=1}^{N}\frac{{\left(B_{i}^{meas}-B_{i}^{model}\right)}^{2}}{N}}$$
where $B_{i}^{meas}$ and $B_{i}^{model}$ are the measured and modelled values for the B-field at each evaluation point, and *N* is the number of measured points.

In this work, the implementation for differential evolution found within the SciPy library [34] was used. Further information on this method can be found in [35,36].

This approach was generally found to work well; however, fitting typically took significantly longer than for the standard five-parameter model due to the increased search space for this fifteen-parameter model.

The minimisation is susceptible to getting trapped in local minima since several combinations of parameters can produce similar results. As such, it can be difficult to be sure the true global minimum has been found. Careful tuning of the parameters of the minimiser goes some way to alleviating this problem; however, it was found that manual selection of a solution from a number of solutions was required to obtain the best result.

## 5. Results and Discussion

The results of the experimental measurements and the proposed modified J–A model are detailed below.

### 5.1. Experimental Results

The obtained major B-H curves from the measurements detailed above are shown in Figure 2; the key features of these curves are discussed below.

#### 5.1.1. As Quenched Samples

The as-quenched samples, A1 and A2, both had similar major loops with a distinct shape from the other samples. Key features were a higher coercivity and lower remanence, along with soft corners on the loops and a very long, shallow approach to saturation at higher applied fields.

#### 5.1.2. Normalised Samples

The normalised samples, D1 and D2 exhibit major loops generally in line with the sigmoid-shaped B-H loop typically seen in soft magnetic materials (and used as the basis for the J–A hysteresis model).

#### 5.1.3. Quenched and Tempered Samples

The B-H loops for the quenched and tempered samples, B1, C1, B2, and C2, were characterised by square corners and a high remanence. These corners were found to be sharper on the B samples.

### 5.2. Curve Fitting

The B-H curves measured above were fitted using both the standard J–A model and the proposed modified J–A model. The results for an as-quenched sample and a quenched and tempered sample are shown below to illustrate the limitations of the standard model and the efficacy of the modified model for representing these types of B-H curves.

### 5.3. As Quenched Steels

In Figure 3, the measured B-H curve for the quenched samples is shown alongside curves modelled with the standard J–A model and the proposed modified model. The measured curves feature a wide central section that then narrows rapidly before slowly and almost reversibly approaching saturation at high fields. This rapid narrowing is not well represented by the standard model. The modified model does not exhibit this problem, and it can be seen that it produces a very good fit to the measured curves.

From Equation (Equation 13), it was found that the total error in the fit of the modified model was 9.6% and 10.3% of that obtained using the standard J–A model for samples A1 and A2, respectively.

### 5.4. Quenched and Tempered Steels

In Figure 4, the measured B-H curve for the quenched and tempered samples is shown alongside curves modelled with the standard J–A model and the proposed modified model.

Considering the measured curves and the curves modelled using the standard model, it can be seen that the measured curve features square corners and, at high fields, a long, almost reversible approach to saturation. Neither of these features is well represented by the standard model, which exhibits rounder corners and flattens too quickly. Again, it can be seen that the modified model addresses these issues and results in very good fits to the measured curves.

The total fitting error of the modified model relative to the standard J–A model was 20.4%, 13.4%, 21.7% and 17.9% for samples B1, B2, C1, and C2, respectively.

## 6. Conclusions

This work has demonstrated the limitations of the standard J–A model for representing the major B-H curves of common engineering steels, particularly those in the as-quenched and quenched-and-tempered states, which exhibit features not well represented by the standard J–A model. These features include the rapid narrowing of the B-H curves for quenched steels and the square corners seen in B-H curves for quenched and tempered steels. A modified model applying Gaussian variations to the model parameters has been proposed in order to address these limitations. By applying the standard and modified models to a selection of experimental measurements, the superior performance of the model has been demonstrated.

Further work on this model could include the application of the model to a wider range of materials; the development of a vector form of the model; extension to minor B-H curves or further development of the form of the equations used for the variation of the model parameters. Possible alternatives to the Gaussian functions proposed include the use of super-Gaussian or bimodal functions.

## Figures and Tables

**Figure 1 sensors-25-01328-f001:**
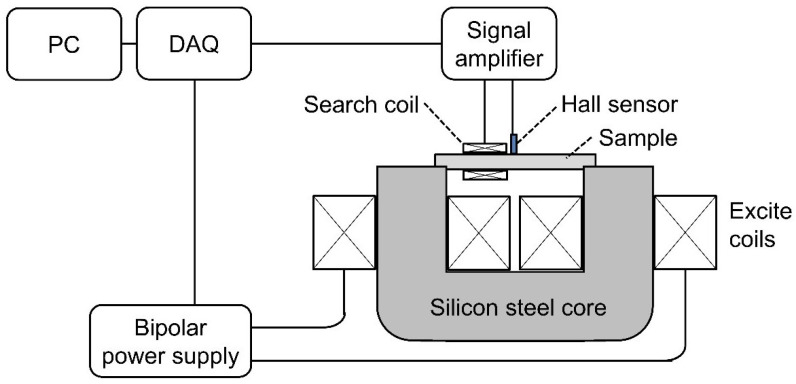
Closed-Circuit B-H Measurement System.

**Figure 2 sensors-25-01328-f002:**
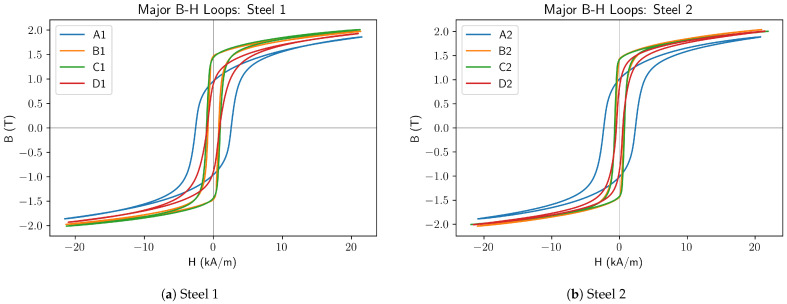
Measured B-H Loops for Cr-Mo (**a**) and C-Mn (**b**) samples detailed in Table 1.

**Figure 3 sensors-25-01328-f003:**
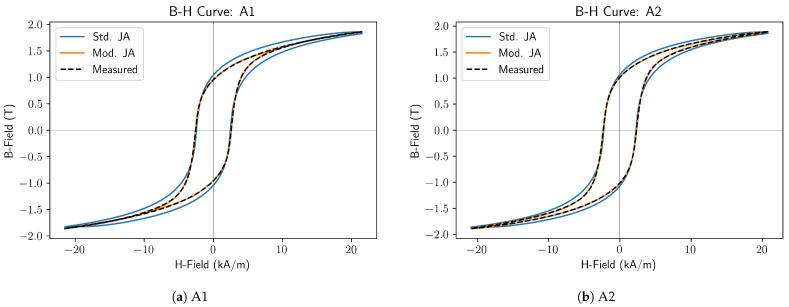
Modified Jiles-Atherton Model: As Quenched Steels.

**Figure 4 sensors-25-01328-f004:**
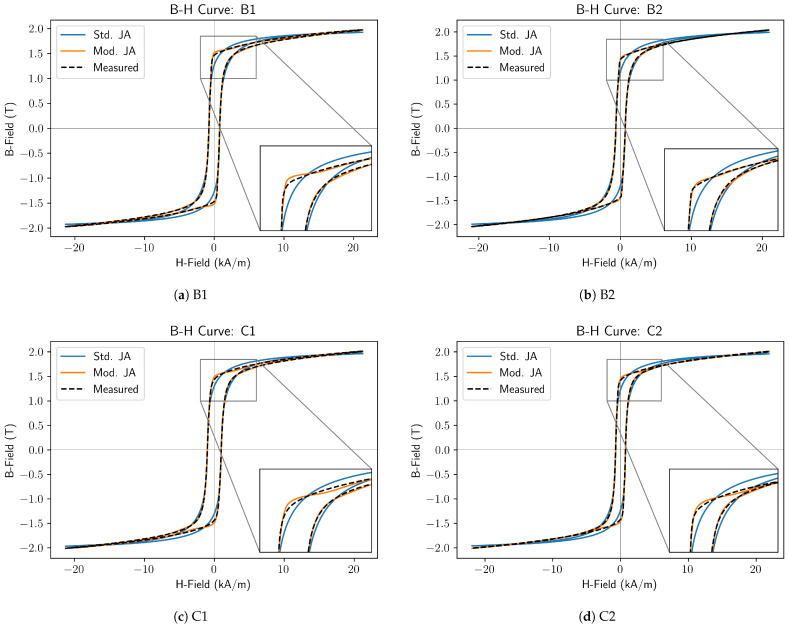
Modified Jiles–Atherton Model: Quenched and Tempered Steels.

**Table 1 sensors-25-01328-t001:** Sample Details.

Sample	Chemistry	Heat Treatment	Average Hardness
A1	Cr-Mo	As quenched	46.5 HRC
B1		Quenched & tempered at 720 °C (30 min)	19 HRC
C1		Quenched & tempered at 590 °C (30 min)	32.3 HRC
D1		Normalised	17.5 HRC
A2	*C*-Mn	As quenched	48.32 HRC
B2		Quenched & tempered at 670 °C (12 min)	18.5 HRC
C2		Quenched & tempered at 500 °C (12 min)	29 HRC
D2		Normalised	85.5 HRB

## Data Availability

The data supporting this study are available from the corresponding authors upon reasonable request.

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
