# Peer review of "Extension to the Jiles–Atherton Hysteresis Model Using Gaussian Distributed Parameters for Quenched and Tempered Engineering Steels"

_sensors, 2025, doi:10.3390/s25051328_

Round 1

Reviewer 1 Report

Comments and Suggestions for Authors

In this paper, the author extended the Jiles-Atherton model for describing the hysteresis of ferromagnetic materials. Gaussian variations were introduced to the proposed model, which is capable of representing the features in the hysteresis curves of quenched and tempered steels. However, the keys reasons why the proposed model gives better fitting is not clearly explained. The author should provide detailed explanation on the reason for choosing Gaussian type variation and why it works. The corresponding error analysis and complexity of algorithm are also suggested to be included.

Author Response

  1. The keys reasons why the proposed model gives better fitting is not clearly explained. The author should provide detailed explanation on the reason for choosing Gaussian type variation and why it works
    • Section 3.1 has been expanded to incorporate a more detailed explanation behind the use of a gaussian type variation.
  2.  The corresponding error analysis and complexity of algorithm are also suggested to be included.
    • Relative errors for curve fitting have been added in sections 5.3 and 5.4
    • Details of the implementation of differential evolution used and additional references on the method have been added in 4.3.

Reviewer 2 Report

Comments and Suggestions for Authors Hysteresis is the most important characteristic of ferri- and ferro-magnetic materials. The authors of this manuscript has made a significant contribution by modifying the Jiles-Atherton model to fit hysteresis loops of steels with varying heat treatments. The presentation needs minor revision. Comparisons in Figures 3 and 4 are unclear. The author should consider resizing figures or adding insets or inserting parameters from the fitting. The manuscript is well written. There are only some inconsistencies in the reference format (e.g. the use of capital letters in journal name) and the use of  Jiles-Atherton/ J-A in the text (the full name should be used once in the introduction)  I recommend the publication of the manuscript. If the journal has issues about the length of the paper (too short) or the lack of up-to-date reference, it may be accepted as Communications/Short Article.

Author Response

  1. The presentation needs minor revision. Comparisons in Figures 3 and 4 are unclear. The author should consider resizing figures or adding insets or inserting parameters from the fitting.
    • Figures have been revised to fill full page width and the font size has been increased. Insets have been added in figure 4.
  2. There are only some inconsistencies in the reference format (e.g. the use of capital letters in journal name)
    • References have been revised to remove inconsistencies.
  3.  the use of  Jiles-Atherton/ J-A in the text (the full name should be used once in the introduction)
    • Added the full name in the introduction.

Reviewer 3 Report

Comments and Suggestions for Authors

This work is devoted to the application of the J-A model to hysteresis curves obtained from experimental measurements for quenched and tempered engineering steels. It has been demonstrated that the current form of the J-A model is not capable of representing certain observed features in the obtained hysteresis curves of these steels. This work has shown that a superior fit to the major loops for such steels can be obtained by applying Gaussian variations with respect to the applied magnetic field to the model parameters. Authors claim that the findings are supported with experimental results. 

- The draft looks more like a technical report of an engineer's work. The text is unclear and does not sound scientific. 

Therefore, it should be shaped out to be more like a scientific paper with a proper structure. 

- No comparison to other works; no comparison to other models: cons and advantages; 

- Details on the computational approach should be provided in the separate section "Methods": grid density, solvers, thresholds, etc. 

- Font style and size on pictures are incoherent throughout the whole paper. In some cases, the font size is so small that it is hard to read text, and in some cases it is too big. 

With the current criticism, I do not recommend publication of this paper.

Author Response

  1. The draft looks more like a technical report of an engineer's work. The text is unclear and does not sound scientific. Therefore, it should be shaped out to be more like a scientific paper with a proper structure. 
    • The paper has been restructured to conform more closely to the MDPI structure with Introduction, Materials and Methods and Results sections. We have retained sections 2 and 3 discussing the Jiles-Atherton Model as we felt this was an appropriate sequence in which to present the material.
  2. No comparison to other works; no comparison to other models: cons and advantages.
    • Relative errors comparing the proposed method to the standard Jiles-Atherton method have now been provided in sections 5.3 and 5.4. As this work is focused on the Jiles-Atherton model we feel a more comprehensive review of the wide range of hysteresis models in the literature would be out of the scope of this paper.
    • A discussion of some other relevant methods is provided in section 3.
  3. Details on the computational approach should be provided in the separate section "Methods": grid density, solvers, thresholds, etc. 
    • Details of computational methods have been moved to section 4.3 within the methods and materials section. This has been expanded slightly to include details of the implementation used and additional references.
  4. Font style and size on pictures are incoherent throughout the whole paper. In some cases, the font size is so small that it is hard to read text, and in some cases it is too big. 
    • Font size has been increased and figures have been expanded to fill the full width of the page. Insets have been added in figure 4 and the font style has been checked for consistency.

Reviewer 4 Report

Comments and Suggestions for Authors

The manuscript describes the results of valuable work on extending the Jilles-Atherton model, which improved the modelling quality of hysteresis loops measured for quenched and tempered engineering steel. The idea of introducing the Gaussian distributions of parameters in the previous model appeared to be clever and effective.

In my opinion, the article will be applicable for publication in the MDPI Sensors journal after minor but obligatory revision according to the remarks given below.

The model works fairly well, however, Authors should present a more pronounced explanation of the physical meaning of their approach. In particular, Authors call the formulas (6)-(10) as Gaussian distributions, whereas they have nothing to do with probability. It would be more adequate to name them as Gaussian-type field dependencies. Of course, in reality, the model parameters may be subject to statistical distributions (around the mean value) due to the inhomogeneity of the system, but the Authors do not consider such a case.

The Authors applied a dedicated AC low-frequency system for quasi-static measurements of magnetization curves. A first glance at the scheme suggests that it is not entirely a closed circuit geometry case. That is why the Authors should explain in the text, how the system was calibrated. In particular, it is essential to explain how “external” magnetic field H was measured with a Hall sensor (was it done in the absence of a sample?).

The caption for Figure 2 is too short and is not informative enough.

In the paper, the table with fitting parameters' values is missing (including those related to the “distributions”).

The list of references is large, up-to-date and of proper choice.

Apart from a few typos, the text is error-free, as it was written by native speakers.

Author Response

  1. Authors should present a more pronounced explanation of the physical meaning of their approach. In particular, Authors call the formulas (6)-(10) as Gaussian distributions, whereas they have nothing to do with probability. It would be more adequate to name them as Gaussian-type field dependencies. Of course, in reality, the model parameters may be subject to statistical distributions (around the mean value) due to the inhomogeneity of the system, but the Authors do not consider such a case.
    • Section 3.1 has been expanded to provide more background as to why gaussian functions have been used and further discuss the physical considerations behind this choice.
    • The model is not intended to reflect probabilistic variations in the model parameters. As such all references to the gaussian distributions have been changed to gaussian functions to avoid confusion.
  2. The Authors applied a dedicated AC low-frequency system for quasi-static measurements of magnetization curves. A first glance at the scheme suggests that it is not entirely a closed circuit geometry case. That is why the Authors should explain in the text, how the system was calibrated. In particular, it is essential to explain how “external” magnetic field H was measured with a Hall sensor (was it done in the absence of a sample?).
    • More details as to how H was measured has been added to section 4.2.
    • The system has not been formally calibrated and it is noted that the system can not be considered as entirely closed circuit. An additional paragraph has been added at the end of section 4.2 discussing this.
  3. The caption for Figure 2 is too short and is not informative enough.
    • Caption changed to: "Measured B-H Loops for Cr-Mo (a) and C-Mn (b) samples detailed in Table 1."
  4. In the paper, the table with fitting parameters' values is missing (including those related to the “distributions”).
    • This has been discussed amongst the authors several times during the initial draft of the paper and during review. Ultimately we have opted not to include this since several combinations of parameters can produce similar results  and due to a reluctance to include the resultant large table of values; a total of 120 values arising from 5 per sample for the standard model and an additional 15 per sample for the modified model.

Round 2

Reviewer 1 Report

Comments and Suggestions for Authors

The author has addressed all the raised questions. I have no more comments.

Reviewer 3 Report

Comments and Suggestions for Authors

I think paper is ready to go.